# Fast Training Set Size Reduction Using Simple Space Partitioning Algorithms

**Stefanos Ougiaroglou** [1] , **Theodoros Mastromanolis** [1], **Georgios Evangelidis** [2] **and Dionisis Margaris** [3,*]

1 Department of Information and Electronic Engineering, School of Engineering, International Hellenic University, 57400 Thessaloniki , Greece
2 Department of Applied Informatics, School of Information Sciences, University of Macedonia, 156 Egnatia Street, 54636 Thessaloniki, Greece
3 Department of Digital Systems, University of the Peloponnese, Valioti's Building, 23100 Sparta, Greece
* Correspondence: margaris@uop.gr

**Abstract:** The Reduction by Space Partitioning (RSP3) algorithm is a well-known data reduction technique. It summarizes the training data and generates representative prototypes. Its goal is to reduce the computational cost of an instance-based classifier without penalty in accuracy. The algorithm keeps on dividing the initial training data into subsets until all of them become homogeneous, i.e., they contain instances of the same class. To divide a non-homogeneous subset, the algorithm computes its two furthest instances and assigns all instances to their closest furthest instance. This is a very expensive computational task, since all distances among the instances of a non-homogeneous subset must be calculated. Moreover, noise in the training data leads to a large number of small homogeneous subsets, many of which have only one instance. These instances are probably noise, but the algorithm mistakenly generates prototypes for these subsets. This paper proposes simple and fast variations of RSP3 that avoid the computationally costly partitioning tasks and remove the noisy training instances. The experimental study conducted on sixteen datasets and the corresponding statistical tests show that the proposed variations of the algorithm are much faster and achieve higher reduction rates than the conventional RSP3 without negatively affecting the accuracy.

**Keywords:** data reduction; Reduction by Space Partitioning; RSP3; prototype generation; instance-based classification; kNN classifier



## 1. Introduction

Data reduction is a crucial pre-processing task [1] in instance-based classification [2]. Its goal is to reduce the high computational cost involved in such classifiers by reducing the training data as much as possible without penalty in classification accuracy. In effect, Data Reduction Techniques (DRTs) attempt to either select or generate a small set of training prototypes that represent the initial large training set so that the computational cost of the classifier is vastly reduced. The selected or generated set of training prototypes is called a condensing set.

DRTs can be based on either the concept of Prototype Selection (PS) [3] or the concept of Prototype Generation (PG) [4]. A PS algorithm collects representative prototypes from the initial training set, while a PG algorithm summarizes similar training instances and generates a prototype that represents them. PS and PG algorithms are based on the hypothesis that training instances far from the boundaries of the different classes, also called class decision boundaries, can be removed without penalty in classification accuracy. On the other hand, the training instances that are close to class decision boundaries are the only useful training instances in instance-based classification. In this paper, we focus on the PG algorithms.

The RSP3 algorithm [5] is a well-known parameter-free PG algorithm. Its condensing set leads to accurate and fast classifiers. However, the algorithm requires high computa-

tional cost to generate its condensing set because it is based on a recursive partitioning process that divides the training set into subsets that contain training instances of only one class, i.e., they are homogeneous. The algorithm keeps dividing each non-homogeneous subset into two new subsets and stops when all created subsets become homogeneous. The center/mean of each subset constitutes a representative prototype that replaces all instances of the subset. In each algorithm step, a subset is divided by finding the pair of its furthest instances. The instances of the initial subset are distributed to the two subsets according to their distances from those furthest instances. The pair of the furthest instances is retrieved by computing all the distances between instances and finding the pair of instances with the maximum distance. The computational cost of this task is high and may become even prohibitive in cases of very large datasets. This weak point constitutes the first motive of the present paper, namely Motive-A.

The quality and the size of the condensing set created by RSP3 depends on the degree of noise in the training data [6]. Suppose that a training instance $x$ that belongs to class $A$ lies in the middle of a data neighborhood with instances that belong to class $B$. In this case, $x$ constitutes noise. RSP3 splits the neighborhood into multiple subsets, with one of them containing only instance $x$. The algorithm mistakenly considers $x$ as a prototype and places it in the condensing set. This observation constitutes the second motive of the present work, namely Motive-B.

In this paper, we propose simple RSP3 variations which consider the two motives presented above. More specifically, this paper proposes:

- RSP3 variations that replace the costly tasks of retrieving the pair of the furthest instances by applying simpler and faster tasks based on which each subset is divided.
- A mechanism for noise removal. This mechanism considers each subset containing only one instance as noise and does not generate prototypes for that subset. As a result, it improves the reduction rates and the classification accuracy when it is applied on noisy training sets. The proposed mechanism can be incorporated in any of the RSP3 variations (conventional RSP3 included).

The experiments show that the proposed variations are much faster than the original version of RSP3. In most cases, accuracy is retained high, and the variations that incorporate the mechanism for noise removal improve the reduction rates and the classification accuracy, especially on noisy datasets. The experimental results are statistically validated by utilizing the Wilcoxon signed rank test and the Friedman test.

The rest of the paper is organized as follows: The recent research works in the field of PG algorithms are reviewed in Section 2. The RSPE algorithm is presented in Section 3. The new RSP3 variations are presented in detail in Section 4. Section 5 presents and discusses the experimental results and the results of the statistical tests. Section 6 concludes the paper and outlines future work.

## 2. Related Work

Prototype Generation algorithms is a research field that has attracted numerous works over the last decades; nowadays, this field is highly active and challenging due to the explosion of Big Data.

In this direction, Triguero et al. [4] review PG algorithms introduced prior to 2012 and present a taxonomy and an experimental comparison of them which show that the RSP3 algorithm achieves considerably high accuracy. Hence, in this paper, we focus our review on research papers published after 2013.

Giorginis et al. [7] introduce two RSP3 variants that accelerate the original RSP3 algorithm by adopting computational geometry methods. The first variant exploits the concept of convex hull in the procedure of finding the pair of the furthest instances in each subset created by RSP3. More specifically, the variant finds the instances that define the convex hull in each subset. Then, it computes only the distances between the convex hull instances and keeps the pair of instances with the largest distance. The second variant is even faster since it approximates the convex hull by finding the Minimum Bounding Rectangle (MBR). The

two variants share the motive of the high computational cost of RSP3 with the algorithms presented in this work. However, the algorithms proposed by the present work avoid complicated computational geometry methods. In effect, the development of the algorithms presented here was motivated by the research conducted in [7].

A fast PG algorithm that is based on k-means clustering [8,9] is Reduction through Homogeneous Clustering (RHC) [10]. Like RSP3, this algorithm is based on the concept of homogeneity. Initially, the algorithm considers the whole training set as a non-homogeneous cluster, and a mean instance is computed for each class present in the cluster. Then, the k-means clustering algorithm uses the aforementioned instances as initial seeds. This procedure is recursively applied for each non-homogeneous discovered cluster until all clusters become homogeneous. The set of means of the homogeneous clusters becomes the final condensing set. The experimental results show that the RHC algorithm is slightly less accurate than RSP3; however, it achieves higher reduction rates. Not only was the RHC algorithm was found to be much faster than RSP3, but it also was one of the fastest approaches that took part in this experimental study [10]. A modified version of the RHC algorithm has recently been applied on string data spaces [11,12].

ERHC [13] is a simple variation of RHC. It considers the clusters that contain only one instance as noise and does not generate prototypes for them. In other words, ERHC incorporates an editing mechanism that removes noise from the training data in its data reduction procedure. New RSP3 variations presented in the present paper adopt the same idea.

Gallego et al. [14] present a simple clustering-based algorithm which accelerates the k-NN classification. Firstly, by using c-means clustering, their algorithm discovers clusters in the training set, the number of which is defined by the user as an input parameter. Afterwards, by searching for the nearest neighbors in the nearest cluster, the k-NN classifier performs classification. Furthermore, the authors use Neural Codes (NC), which are feature-based representations extracted by Deep Neural Networks, in order to further improve their algorithm. These NC collect same class instances in order to be placed within the same cluster. Typically, this algorithm cannot be considered a PG algorithm. Although the paper refers to the means of clusters as prototypes, the algorithm does not achieve training data reduction. However, the authors empirically compare their algorithm against several DRTs.

The algorithms presented in [15,16] cannot be considered as PG. Both are based on pre-processing tasks that build a two-level data structure. The first level holds prototypes while the second one stores the "real" training instances. The classification is performed by accessing either the first or the second level of the data structure. The decision is based on the area where the new unclassified instance lies and according to pre-specified criteria. Like the previous paper, the authors compare the algorithms against DRTs.

The algorithm presented in [17] cannot be considered a PG algorithm. However, it is able to perform efficient Nearest Neighbor searches in the context of $k$-NN classification. As the authors state, the proposed caKD+ algorithm combines clustering, feature selection, different k parameters in each resulted cluster and multi-attribute indexing in order to perform efficient $k$-NN searches and classification.

Impedovo et al. [18] introduce a handwriting digit recognition PG algorithm. This algorithm consists of two phases. In the first one, using the Adaptive Resonance Theory [19], the number of prototypes is determined, and the initial set of prototypes is generated. In the second phase, a naive evolution strategy is used to generate the final set of prototypes. The technique is incremental, and, by modifying the previously generated prototypes or by adding new prototypes, it can be adapted to writing style changes.

Rezaei and Nezamabadi-pour [20] present a swarm-based metaheuristic search algorithm inspired by motion and gravity Newtonian laws [21], namely the gravitational search algorithm, which is adapted for prototype generation. The authors include the RSP3 in their experimental study.

Hu and Tan [22] improve the performance of NN classification by presenting two methods for evolutionary computation-based prototype generation. The first one, namely error rank, targets at upgrading the NN classifier's generation ability by taking into account

the misclassified instances, while the second one is able to avoid over-fitting by pursuing the performance on multiple data subsets. The paper shows that by using the two proposed methods, particle swarm optimization achieves better classification performance. This paper also includes RSP3 in the experimental study.

Elkano et al. [23] present an one-pass Map-Reduce Prototype Generation technique, namely CHI-PG, which exploits the Map-Reduce paradigm and uses fuzzy rules in order to produce prototypes that are exactly the same, regardless of the number of Mappers/Reducers used. The proposed approach enhances the distributed prototype reduction execution time without decreasing classification reduction rates and accuracy; however, its input parameters must be empirically determined.

Escalante et al. [24] present a PG algorithm, namely Prototype Generation via Genetic Programming, which is based on genetic programming. This algorithm generates prototypes that maximize an estimate of the NN classifier's generalization performance by combining training instances through arithmetic operators. Furthermore, the proposed algorithm is able to automatically select the number of prototypes for each class.

Calvo-Zaragoza et al. [25] use dissimilarity space techniques in order to apply PG algorithms to structured representations. The initial structural representation is mapped to a feature-based one, hence allowing the use of statistical PG methods on the original space. In the experimental study, RSP3 and two other PG algorithms are used, while the results show that RSP3 achieves the highest accuracy.

Cruz-Vega and Escalante [26] present a Learning Vector quantization technique, which is based on granular computing and includes Big Data incremental learning mechanisms. This technique, firstly, groups instances with similar features very fast, by using a one-pass clustering task, and, then, it covers the class distribution by generating prototypes. It comprises two stages. In the first one, the number of prototypes is controlled using a usage-frequency indicator, and the best prototype is kept using a life index, while in the second one, the useless dimensions are pruned of the training database.

Escalante et al. [27] present a prototype generation multi-objective evolutionary technique. This technique targets enhancing the reduction rate and accuracy at the same time, and achieving a better trade-off between them, by formulating the prototype generation task as a multi-objective optimization problem. In this technique, the amount of prototypes, as well as the generalization performance estimation that the selected prototypes achieve, are the key factors. The authors include RSP3 in their experimental study.

The algorithm proposed by Brijnesh J. Jain and David Schultz in [28] adapts the Learning Vector Quantization (LVQ) PG method in time-series classification. In effect, the paper extends the LVQ approach from Euclidean spaces to Dynamic Time Wrapping spaces. The work presented by Leonardo A. Silva et al. [29] focuses on the number of prototypes generated by PG algorithms. The work introduces a model that estimates the ideal number of prototypes according to the characteristics of the dataset used. Last but not least, I. Sucholutsky and M. Schonlau in [30] focus on PG methods for datasets with complex geometries.

## 3. The Original Rsp3 Algorithm

RSP3 is one of the three proposed RSP algorithms [5]. The three algorithms are descendants of the Chen and Jozwik algorithm (CJA) [31]. However, RSP3 is the only parameter-free RSP algorithm (CJA included) and builds the same condensing set regardless of the order of data in the training set.

RSP3 works as follows: It initially finds the pair of the furthest instances, $a$ and $b$, in the training set (see Figure 1). Then, it splits the training set into two subsets, $C_a$ and $C_b$, with the training instances assigned to their closest furthest instance. Then, in each algorithm iteration and by following the aforementioned procedure, a non-homogeneous subset is divided into two subsets. The splitting tasks stop when all created subsets become homogeneous. Then, the algorithm generates prototypes. For each created subset $C$, RSP3 computes its mean by averaging its training instances. The mean instance that is labeled by

the class of the instances in *C* plays the role of a generated prototype and is placed in the condensing set.

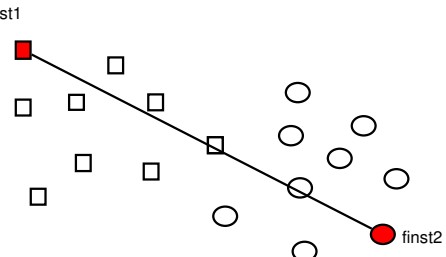

finst1

finst2

**Figure 1.** The RSP3 algorithm divides the subset according to its furthest instances *finst*1 and *finst*2.

The pseudo-code presented in Algorithm 1 is a possible non-recursive implementation of RSP3 that uses a data structure *S* to store subsets. In the beginning, the whole training set (*TS*) is a subset to be processed, and it is placed in *S* (line 2). At each iteration, RSP3 selects a subset *C* from *S* and checks whether it is homogeneous or not. If *C* is homogeneous, the algorithm computes its mean instance and stores it in the condensing set (*CS*) as a prototype (lines 6–9). Then, *C* is removed from *S* (line 17). If *C* is non homogeneous, the algorithm finds the furthest instances *a* and *b* in *C* (line 11) and divides *C* into two subsets $C_a$ and $C_b$ by assigning each instance of *C* to its closest furthest instance (lines 12–13). The new subsets $C_a$ and $C_b$ are added to *S* (lines 14–15), and *C* is removed from *S* (line 17). The loop terminates when *S* becomes empty (line 18), i.e., when all subsets become homogeneous.

---

**Algorithm 1** RSP3

---

**Input:** *TS* {Training Set}
**Output:** *CS* {Condensing Set}
 1: $S \leftarrow \varnothing$ {Initialize structure that holds unprocessed subsets}
 2: add(*S*, *TS*)
 3: $CS \leftarrow \varnothing$ {Initialize *CS*}
 4: **repeat**
 5:     $C \leftarrow$ pick a subset from *S*
 6:     **if** *C* is homogeneous **then**
 7:         $r \leftarrow$ mean instance of *C*
 8:         *r.label* $\leftarrow$ class of instances in *C*
 9:         $CS \leftarrow CS \cup \{r\}$ {add *r* in the condensing set}
10:     **else**
11:         $(a, b) \leftarrow$ furthest instances in *C* {Algorithm 2 is applied}
12:         $C_a \leftarrow$ set of *C* instances closer to *a*
13:         $C_b \leftarrow$ set of *C* instances closer to *b*
14:         add(*S*, $C_a$)
15:         add(*S*, $C_b$)
16:     **end if**
17:     remove(*S*, *C*)
18: **until** IsEmpty(*S*) {all subsets became homogeneous}
19: **return** *CS*

---

In the close to class decision boundaries areas, the training instances from different classes are close to each other. RSP3 creates more prototypes for those data areas, since many small homogeneous subsets are created. Similarly, more subsets are created and more prototypes are generated for noisy data areas. In effect, a subset with only one instance constitutes noise. In contrast, fewer and larger subsets are created for the "internal" data areas which are far from the decision boundaries where a class dominates.

Sanchez, in his experimental study presented in [5], showed that RSP3 generates a small condensing set. When an instance-based classifier such as *k*-NN utilizes the RSP3 generated condensing set, it achieves accuracy almost as high as when *k*-NN runs over

the original training set. However, the computational cost of the classification step is significantly lower.

The retrieval of the pair of the furthest instances in each subset requires the computation of all distances between the instances of the subset. This approach is simple and straightforward. However, it is a computationally expensive task that burdens the overall pre-processing cost of the algorithm. In cases of large datasets, this drawback may render the execution of RSP3 prohibitive.

In this respect, the conventional RSP3 algorithm computes $\frac{|C| \times (|C|-1)}{2}$ distances in order to find the most distant instances in each subset $C$. Thus, for each subset division, RSP3 proceeds with the pseudo-code outlined in Algorithm 2.

---

**Algorithm 2** The Grid algorithm

---

**Input:** $C$ {A subset containing instances $inst_1$ through $inst_{|C|}$}
**Output:** $D_{max}, finst_1, finst_2$

 1: $D_{max} \leftarrow 0$
 2: **for** $i \leftarrow 1$ to $|C|$ **do**
 3:    **for** $j \leftarrow i + 1$ to $|C|$ **do**
 4:       $D_{curr} \leftarrow distance(inst_i, inst_j)$
 5:       **if** $D_{curr} > D_{max}$ **then**
 6:          $D_{max} \leftarrow D_{curr}$
 7:          $finst_1 \leftarrow inst_i$
 8:          $finst_2 \leftarrow inst_j$
 9:       **end if**
10:    **end for**
11: **end for**
12: **return** $D_{max}, finst_1, finst_2$

---

In effect, a grid of distances is computed; hence, Algorithm 2 is labeled "The Grid Algorithm". It returns the furthest instances $finst_1$ and $finst_2$ in $C$ along with their distance $D_{max}$. Hereinafter, each reference to the "RSP3" acronym implies the RSP3 algorithm whereby the most distant instances are calculated by applying the Grid algorithm to all the instances in each subset. It is worth mentioning that RSP3 as implemented in the KEEL software [32] applies this simple and straightforward approach for finding the pair of the most distant instances in $C$.

## 4. The Proposed Rsp3 Variations

### 4.1. The Rsp3 with Editing (Rsp3e) Algorithm

The RSP3 with editing (RSP3E) algorithm incorporates an editing mechanism that removes noise from the training data. RSP3E is almost identical to the conventional RSP3. However, it involves a major difference: If a subset with only one instance is created, this subset is considered to be noise. In effect, such an instance is surrounded by instances that belong to different classes. The algorithm does not proceed with the prototype generation for this subset. Therefore, for each subset containing only one instance, RSP3E does not generate a prototype in the condensed set. RSP3E addresses Motive-B (defined in Section 1). RSP3E has been inspired by the idea first adopted by ERHC [13] and EHC [33].

### 4.2. The Rsp3-Rnd and Rsp3e-Rnd Algorithms

As already explained, RSP3 finds the pair of the furthest instances in each subset in order to divide it. Likely, the most distant instances in a subset belong to different classes. By splitting the subset using such instances, the probability of creating two large homogeneous subsets is higher. Thus, RSP3 may need fewer iterations in order to divide the whole training set into homogeneous subsets, and the reduction rates may be higher.

The RSP3-RND and RSP3E-RND algorithms were inspired by the following observation: RSP3 can run and produce a condensing set even if it selects any pair of instances instead of the pair of the furthest instances. In that case, the algorithm will likely need

more subset divisions and the data reduction rate will be lower. However, the procedure of subset division will be much faster, since the costly retrieval of the furthest instances will be avoided. This simple idea is adopted by RSP3-RND and RSP3E-RND that work similarly to RSP3 and RSP3E, respectively, but they randomly select the pair of instances used for subset division.

RSP3-RND and RSP3E-RND will generate different condensing sets in different executions. In other words, the number of divisions and the generated prototypes depend on the selection of the random pairs of instances. RSP3-RND addresses Motive-A, while RSP3E-RND addresses both motives.

### 4.3. The Rsp3-M and Rsp3e-M Algorithms

The RSP3-M and RSP3E-M algorithms are two more simple variations of RSP3 and RSP3E, respectively. Both work as follows: Initially, the algorithms find the two top classes in terms of instances belonging to them. These classes are called the *common* classes. The mean instances of the common classes constitute the pair of instances based on which a non-homogeneous set is divided into two subsets (see Figure 2).

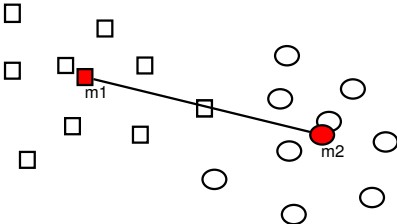

**Figure 2.** The RSP3-M and RSP3E-M algorithms divide a subset according to the means of the two most common classes in the subset $m1$ and $m2$.

Obviously, similar to RPS3-RND and RSP3E-RND, RSP3-M addresses Motive-A while RSP3E-M addresses both motives. In effect, RSP3-M and RSP3E-M speed up the original algorithm because they replace the computation of the furthest instances of a set with the computation of the two common classes of the set and the corresponding mean instances, which is a much faster approach. The idea behind RSP3-M and RSP3E-M is quite simple: By dividing a non-homogeneous set into two subsets based on the means of the most common classes in the subset, it is more probable for the algorithms to earlier obtain large homogeneous subsets. We expect that both RSP3-M and RSP3E-M will increase the reduction rates at a maximum level. However, this may negatively affect accuracy.

### 4.4. The Rsp3-M2 and Rsp3e-M2 Algorithms

The RSP3-M2 and RSP3E-M2 algorithms are almost identical to RSP3-M and RSP3E-M, respectively. The only difference is that instead of using the generated mean instances of the most common classes in order to divide a non-homogeneous subset, RSP3-M2 and RSP3E-M2 identify and use the training instances that are closer to the mean instances (see Figure 3). We expect that this may reduce the reduction rates, and as a result, the accuracy achieved by RSP3-M2 and RSP3E-M2 will be higher compared to that of RSP3-M and RSP3E-M.

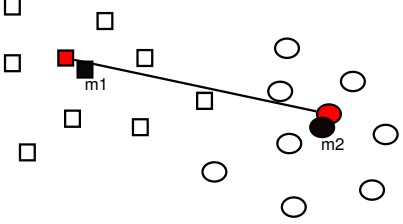

**Figure 3.** The RSP3-M2 and RSP3E-M2 algorithms divide a subset according to the instances that are closer to the means of the two most common classes in the subset $m1$ and $m2$.

## 5. Experimental Study

### 5.1. Experimental Setup

The original RSP3 algorithm and its proposed variations were coded in C++. Moreover, we include the results of the NOP approach (no data reduction) for comparison purposes. The experiments were conducted on a Debian GNU/Linux server equipped with a 12-core CPU with 64 GB of RAM. The experimental results were measured by running the k-NN classifier (with $k = 1$) over the original training set (case of NOP classifier) and the condensing sets generated by the conventional RSP3 algorithm and its proposed variations. The $k$ parameter value is the only parameter used in the experimental study. Following the common practice in the field of data reduction for instance-based classifiers, we used the setting $k = 1$.

We used 16 datasets distributed by the KEEL [34] and UCI machine learning [35] repositories, whose main characteristics are summarized in Table 1. Each dataset's attribute values were normalized to the range [0, 1], and we used the Euclidean distance as a similarity measure. We removed all nominal and fixed-value attributes and the duplicate instances from the KDD dataset, thus reducing its size to 141,481 instances.

As mentioned above, the major goal of the proposed variants of RSP3 is to minimize the computational cost needed for the condensing set construction. High reduction rates as well as keeping the accuracy at high levels are also goals. Thus, for each algorithm and dataset, we used a five-fold cross-validation schema to measure the following four metrics: (i) Accuracy (ACC), (ii) Reduction Rate (RR), (iii) Distance Computations (DC) required for the condensing set construction (in millions (M)), and, (iv) CPU time (CPU) in seconds required for the condensing set construction.

**Table 1.** Datasets characteristics.

| Dataset | Instances | Attributes | Classes |
|---|---|---|---|
| Balance (BL) | 625 | 4 | 3 |
| KDD Cup (KDD) | 494,020/141,481 | 40 | 23 |
| Banana (BN) | 5300 | 2 | 2 |
| Letter Image Recognition (LIR) | 20,000 | 16 | 26 |
| Landsat Satellite (LS) | 6435 | 36 | 7 |
| Magic Gamma Telescope (MGT) | 19,020 | 11 | 2 |
| MONK-2 (MNK) | 432 | 6 | 2 |
| Pen Digits (PD) | 10,992 | 16 | 10 |
| Phoneme (PH) | 5404 | 5 | 2 |
| Shuttle (SH) | 58,000 | 9 | 7 |
| Textrue (TXR) | 5500 | 40 | 11 |
| Yeast (YST) | 1484 | 8 | 10 |
| Pima (PM) | 768 | 8 | 2 |
| Twonorm (TN) | 7400 | 20 | 2 |
| Waveform (WF) | 5000 | 21 | 3 |
| Eye State (EEG) | 14,980 | 14 | 2 |

### 5.2. Experimental Results

Table 2 presents, for each dataset and algorithm, the ACC, RR, DC and CPU measurements. Table 3 summarizes the measurements of Table 2 and presents the average measurements as well as the standard deviation and the coefficient variance of the measurements.

Furthermore, Figures 4–7 present an overview of average measurements in bar diagrams. More specifically, Figure 4 depicts the average accuracy measurements computed by averaging the ACC measurements achieved by the 1-NN classifier using the condensing set generated by the algorithms. Correspondingly, Figure 5 presents the average RR measurements achieved by the algorithms on the different datasets. Figure 6 illustrates the average distance computations and Figure 7 shows the average CPU times. The diagrams presented in Figures 4–6 are in linear scale, while the diagram presented in Figure 7 is in logarithmic scale.

The results reveal that all algorithms are relatively close in terms of accuracy. However, RPS3, RSP3E, RSP3-RND, RSP3E-RND and RSP3E-M2 achieve the highest ACC measurements. Nevertheless, the high reduction rates achieved by RSP3-M, RSP3-M2 and RSP3E-M seem to negatively affect accuracy. Almost in all cases, RSP3E achieves the highest accuracy, while RSP3E-RND and RSP3-M2 follow. The results indicate that the editing mechanism incorporated by these algorithms is effective.

Concerning RR measurements, we observe in Table 2 that RSP3E-M has the highest performance. However, as mentioned above, these high reduction rates negatively affect accuracy. Furthermore, we observe that the algorithms that incorporate the editing mechanism seem to be more effective in terms of RR measurements. In particular, by removing the useless noisy instances from the data, they achieve higher RR measurements than the algorithms that do not incorporate editing and, at the same time, their accuracy is either improved or is not negatively affected.

Moreover, we can observe that the proposed RSP3 variations outperform the original RSP3, in terms of RR, DC and CPU measurements, which concern the pre-processing cost required for the condensing set construction. This happens because RSP3 computes a large number of distances. In contrast, the proposed variations divide the subsets by avoiding computationally costly procedures. As far as the large datasets are concerned (i.e., KDD, SH, LIR, MGT), the gains are extremely high. In contrast, RSP3 leads to noticeably high CPU costs. Figures 6 and 7 visualize this extreme superiority in terms of pre-processing computational cost.

As far as the large datasets are concerned (i.e., KDD, SH), RSP3 leads to noticeably intensive CPU. The experimental results reveal that RSP3-M and RSP3E-M are faster than RSP3-M2 and RSP3E-M2, respectively. In addition, RSP3-M2 and RSP3E-M2 are faster than RSP3-RND and RSP3E-RND, and the latter are faster then the original RSP3 algorithm and the proposed RSP3E variant.

By observing Tables 2 and 3 and Figures 4–7, we observe that the variations with the editing mechanism that removes the subsets containing only one instance (i.e., RSP3E, RSP3E-RND, RSP3E-M and RSP3E-M2) achieve quite higher RR measurements when compared with the corresponding methods without the editing mechanism, and, at the same time, in most cases, they achieve higher accuracy.

Finally, the experimental results show that the RR measurements achieved by RSP3E-M are the highest. In contrast, as expected, RSP3-RND is the algorithm with the lowest reduction rates.

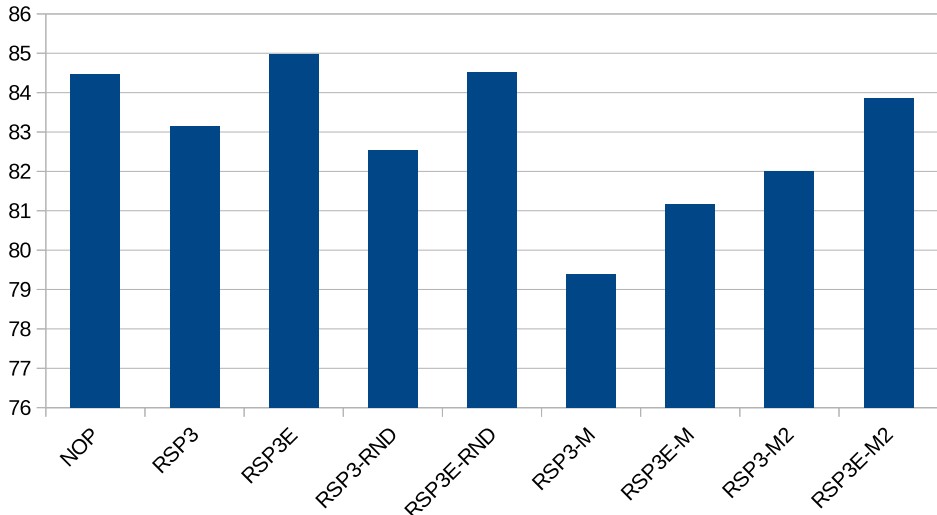

**Figure 4.** Average accuracy measurements.

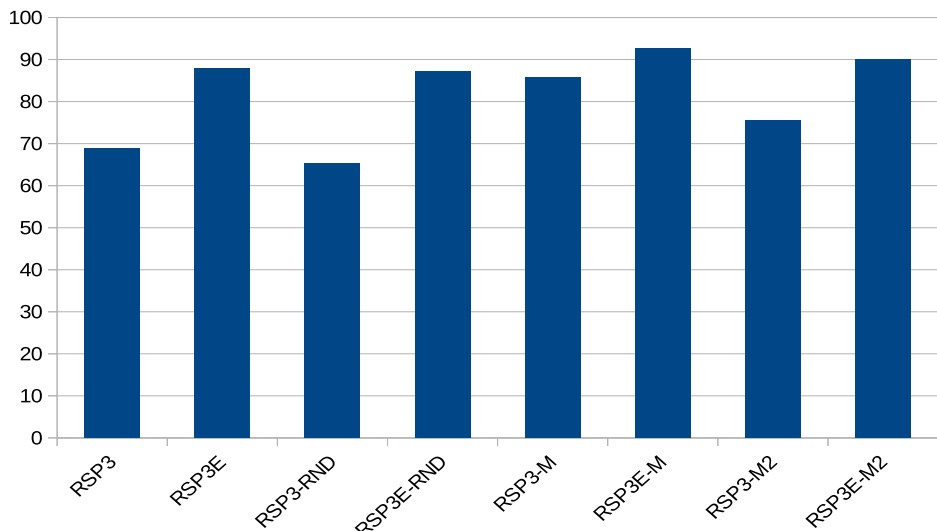

**Figure 5.** Average reduction rates measurements.

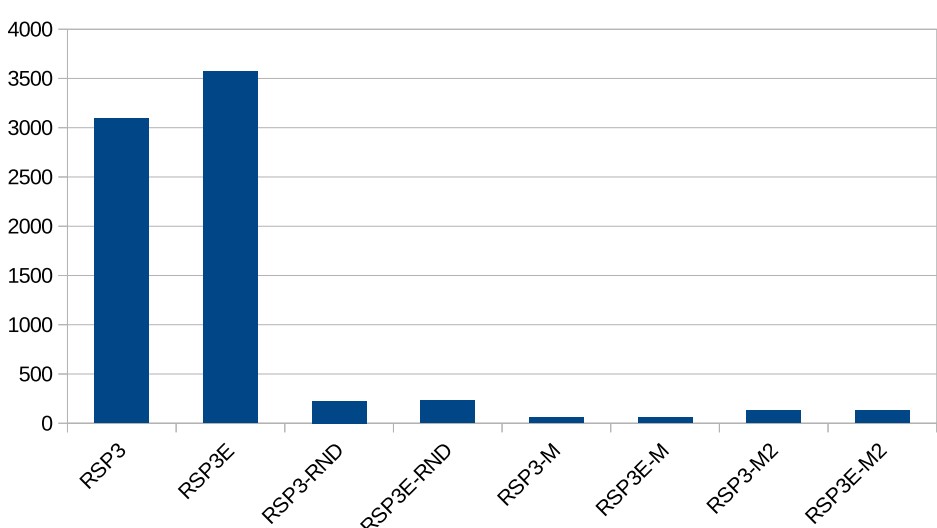

**Figure 6.** Average distance computations (in millions).

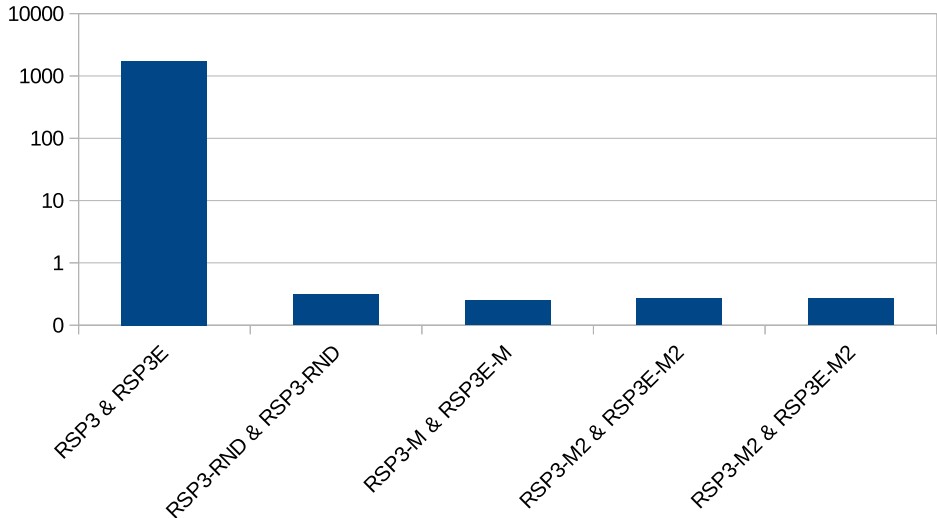

**Figure 7.** Average CPU measurements (in secs).

**Table 2.** Comparison in terms of ACC(%)), RR(%), DC(M) and CPU (Secs).

| Dataset | | NOP | RSP3 | RSP3E | RSP3-RND | RSP3E-RND | RSP3-M | RSP3E-M | RSP3-M2 | RSP3E-M2 |
|---|---|---|---|---|---|---|---|---|---|---|
| BL | ACC: | 80.61 | 72.75 | 86.86 | 74.37 | 86.86 | 66.19 | 71.80 | 69.72 | 82.21 |
| | RR: | - | 63.20 | 86.40 | 58.60 | 86.44 | 83.76 | 91.48 | 70.76 | 89.44 |
| | DC: | - | 0.254 | 0.254 | 0.007 | 0.007 | 0.005 | 0.005 | 0.007 | 0.007 |
| | CPU: | - | 0.198 | 0.149 | 0.087 | 0.062 | 0.039 | 0.042 | 0.068 | 0.089 |
| KDD | ACC: | 99.71 | 99.60 | 99.62 | 99.46 | 99.52 | 98.91 | 98.79 | 98.87 | 99.05 |
| | RR: | - | 98.54 | 99.06 | 97.76 | 98.53 | 99.30 | 99.58 | 99.09 | 99.44 |
| | DC: | - | 20,278.7 | 20,278.7 | 2.1 | 2.1 | 1.8 | 1.8 | 1.8 | 1.8 |
| | CPU: | - | 42,388.5 | 49,080.4 | 703.9 | 670.8 | 163.4 | 153.7 | 170.0 | 177.1 |
| BN | ACC: | 86.92 | 84.60 | 88.36 | 84.43 | 88.30 | 80.61 | 87.81 | 84.47 | 87.51 |
| | RR: | - | 75.09 | 90.43 | 73.70 | 89.83 | 82.39 | 91.73 | 77.96 | 90.43 |
| | DC: | - | 18.88 | 18.88 | 0.078 | 0.078 | 0.066 | 0.066 | 0.077 | 0.077 |
| | CPU: | - | 16.96 | 15.22 | 10.61 | 10.19 | 8.78 | 8.79 | 9.20 | 8.96 |
| LIR | ACC: | 95.75 | 95.56 | 91.87 | 95.56 | 90.98 | 92.05 | 89.82 | 94.70 | 92.03 |
| | RR: | - | 61.88 | 84.08 | 54.31 | 83.89 | 82.74 | 91.30 | 71.62 | 87.66 |
| | DC: | - | 329.8 | 329.8 | 0.48 | 0.48 | 0.39 | 0.39 | 0.44 | 0.44 |
| | CPU: | - | 1065.8 | 1175.6 | 1277.8 | 1418.0 | 199.9 | 189.9 | 539.5 | 541.9 |
| LS | ACC: | 89.94 | 89.76 | 89.06 | 90.02 | 89.79 | 86.62 | 87.02 | 89.46 | 89.70 |
| | RR: | - | 72.89 | 89.06 | 69.38 | 88.15 | 90.48 | 94.17 | 79.03 | 90.94 |
| | DC: | - | 34.02 | 34.02 | 0.11 | 0.11 | 0.08 | 0.08 | 0.11 | 0.11 |
| | CPU: | - | 113.3 | 109.3 | 25.9 | 26.1 | 3.8 | 3.7 | 11.1 | 11.8 |
| MGT | ACC: | 80.50 | 77.41 | 81.98 | 77.32 | 82.04 | 71.98 | 77.11 | 76.99 | 81.46 |
| | RR: | - | 58.75 | 84.32 | 56.48 | 84.37 | 80.68 | 89.11 | 64.50 | 85.34 |
| | DC: | - | 364.8 | 364.8 | 0.41 | 0.41 | 0.32 | 0.32 | 0.43 | 0.43 |
| | CPU: | - | 1062.3 | 1156.0 | 858.8 | 871.5 | 319.9 | 315.6 | 813.9 | 818.1 |
| MNK | ACC: | 90.51 | 91.22 | 91.21 | 80.31 | 84.71 | 88.88 | 89.81 | 87.50 | 87.73 |
| | RR: | - | 61.33 | 81.68 | 71.85 | 78.38 | 95.20 | 95.55 | 95.67 | 95.95 |
| | DC: | - | 0.125 | 0.125 | 0.004 | 0.004 | 0.002 | 0.002 | 0.002 | 0.002 |
| | CPU: | - | 0.098 | 0.099 | 0.017 | 0.017 | 0.007 | 0.007 | 0.007 | 0.007 |
| PD | ACC: | 99.33 | 99.16 | 99.05 | 99.00 | 98.73 | 96.78 | 96.95 | 98.23 | 98.39 |
| | RR: | - | 89.64 | 93.31 | 82.80 | 89.83 | 96.56 | 97.75 | 93.59 | 96.07 |
| | DC: | - | 86.14 | 86.14 | 0.19 | 0.19 | 0.15 | 0.15 | 0.17 | 0.17 |
| | CPU: | - | 133.35 | 122.85 | 29.45 | 25.86 | 2.85 | 2.53 | 6.09 | 5.97 |
| PH | ACC: | 89.64 | 86.62 | 86.16 | 86.21 | 85.90 | 82.73 | 83.79 | 86.21 | 85.80 |
| | RR: | - | 69.31 | 85.67 | 67.40 | 86.42 | 80.94 | 89.29 | 74.08 | 86.95 |
| | DC: | - | 21.37 | 21.37 | 0.09 | 0.09 | 0.07 | 0.07 | 0.09 | 0.09 |
| | CPU: | - | 25.04 | 22.76 | 17.06 | 18.75 | 8.55 | 8.21 | 11.59 | 11.73 |
| SH | ACC: | 99.93 | 99.48 | 99.58 | 99.25 | 99.20 | 99.01 | 98.71 | 95.83 | 95.95 |
| | RR: | - | 99.41 | 99.54 | 98.76 | 98.98 | 99.65 | 99.72 | 99.67 | 99.74 |
| | DC: | - | 5991.0 | 5991.0 | 0.77 | 0.77 | 0.60 | 0.60 | 0.64 | 0.64 |
| | CPU: | - | 3517.9 | 4183.2 | 16.8 | 14.9 | 5.2 | 4.4 | 5.4 | 5.0 |
| TXR | ACC: | 98.91 | 98.62 | 97.65 | 98.35 | 97.58 | 95.87 | 94.85 | 97.42 | 96.69 |
| | RR: | - | 82.32 | 89.84 | 77.92 | 87.81 | 94.62 | 96.41 | 88.94 | 93.67 |
| | DC: | - | 25.74 | 25.74 | 0.09 | 0.09 | 0.06 | 0.06 | 0.07 | 0.07 |
| | CPU: | - | 84.25 | 81.33 | 8.21 | 8.03 | 1.70 | 1.58 | 3.26 | 3.15 |
| YST | ACC: | 51.58 | 49.76 | 56.30 | 50.50 | 54.42 | 45.38 | 50.51 | 47.34 | 53.68 |
| | RR: | - | 28.16 | 83.83 | 25.78 | 84.47 | 50.13 | 81.33 | 33.99 | 82.54 |
| | DC: | - | 2.14 | 2.14 | 0.02 | 0.02 | 0.02 | 0.02 | 0.02 | 0.02 |
| | CPU: | - | 3.87 | 3.33 | 2.38 | 2.14 | 1.47 | 1.36 | 2.14 | 2.14 |
| PM | ACC: | 70.54 | 67.67 | 70.93 | 67.67 | 73.14 | 63.49 | 68.97 | 67.80 | 71.18 |
| | RR: | - | 44.56 | 81.63 | 40.91 | 81.95 | 69.67 | 84.53 | 50.78 | 82.25 |
| | DC: | - | 0.56 | 0.56 | 0.012 | 0.012 | 0.008 | 0.008 | 0.011 | 0.011 |
| | CPU: | - | 0.81 | 0.60 | 0.27 | 0.43 | 0.10 | 0.14 | 0.28 | 0.23 |
| TN | ACC: | 94.7 | 93.11 | 95.61 | 93.51 | 95.55 | 82.84 | 84.65 | 92.68 | 94.84 |
| | RR: | - | 84.31 | 92.21 | 74.17 | 88.48 | 97.91 | 98.52 | 84.35 | 92.59 |
| | DC: | - | 37.49 | 37.49 | 0.15 | 0.15 | 0.05 | 0.05 | 0.12 | 0.12 |
| | CPU: | - | 74.53 | 67.32 | 22.83 | 21.75 | 1.09 | 0.79 | 8.39 | 8.15 |
| WF | ACC: | 77.26 | 77.54 | 81.04 | 77.94 | 80.88 | 70.84 | 71.94 | 77.86 | 80.36 |
| | RR: | - | 57.03 | 85.11 | 50.31 | 84.42 | 91.15 | 93.90 | 61.36 | 86.23 |
| | DC: | - | 16.99 | 16.99 | 0.11 | 0.11 | 0.06 | 0.06 | 0.1 | 0.1 |
| | CPU: | - | 52.95 | 46.95 | 32.09 | 28.85 | 1.81 | 1.49 | 17.79 | 17.66 |
| EEG | ACC: | 45,62 | 47.31 | 44.48 | 46.81 | 44.71 | 48.10 | 46.21 | 46.93 | 45.13 |
| | RR: | - | 53.76 | 81.20 | 45.75 | 82.83 | 76.88 | 86.82 | 61.01 | 82.01 |
| | DC: | - | 499.1 | 499.1 | 0.36 | 0.36 | 0.26 | 0.26 | 0.39 | 0.39 |
| | CPU: | - | 972.4 | 1054.5 | 639.2 | 597.6 | 249.1 | 239.0 | 493.5 | 492.2 |
| AVG | ACC: | 84.47 | 83.14 | 84.99 | 82.54 | 84.52 | 79.39 | 81.17 | 82.00 | 83.86 |
| | RR: | - | 68.76 | 87.96 | 65.37 | 87.17 | 85.75 | 92.57 | 75.40 | 90.08 |
| | DC: | - | 1731.69 | 1731.69 | 0.31 | 0.31 | 0.25 | 0.25 | 0.27 | 0.27 |
| | CPU: | - | 3094.52 | 3569.98 | 227.84 | 232.19 | 60.48 | 58.20 | 130.76 | 131.51 |

**Table 3.** Statistics of experimental measurements (Average (AVG), Standard Deviation (STDEV), Coefficient of Variation (CV)).

| Dataset | | NOP | RSP3 | RSP3E | RSP3-RND | RSP3E-RND | RSP3-M | RSP3E-M | RSP3-M2 | RSP3E-M2 |
|---|---|---|---|---|---|---|---|---|---|---|
| AVG | ACC: | 84.47 | 83.14 | 84.99 | 82.54 | 84.52 | 79.39 | 81.17 | 82.00 | 83.86 |
| | RR: | - | 68.76 | 87.96 | 65.37 | 87.17 | 85.75 | 92.57 | 75.40 | 90.08 |
| | DC: | - | 1731.69 | 1731.69 | 0.31 | 0.31 | 0.25 | 0.25 | 0.27 | 0.27 |
| | CPU: | - | 3094.52 | 3569.98 | 227.84 | 232.19 | 60.48 | 58.20 | 130.76 | 131.51 |
| STD | ACC: | 16.53 | 16.77 | 15.71 | 16.52 | 15.63 | 17.04 | 15.92 | 16.71 | 15.49 |
| | RR: | - | 19.28 | 5.78 | 19.76 | 5.44 | 13.01 | 5.42 | 18.36 | 5.82 |
| | DC: | - | 5161.65 | 5161.65 | 0.52 | 0.52 | 0.45 | 0.45 | 0.45 | 0.45 |
| | CPU: | - | 10,517.29 | 12,182.89 | 403.95 | 425.69 | 107.31 | 104.05 | 252.08 | 253.07 |
| CV | ACC: | 84.47 | 0.20 | 0.20 | 0.18 | 0.20 | 0.18 | 0.21 | 0.20 | 0.18 |
| | RR: | - | 0.28 | 0.07 | 0.30 | 0.06 | 0.15 | 0.06 | 0.24 | 0.06 |
| | DC: | - | 2.98 | 2.98 | 1.68 | 1.68 | 1.82 | 1.82 | 1.67 | 1.67 |
| | CPU: | - | 3.40 | 3.41 | 1.77 | 1.83 | 1.77 | 1.79 | 1.93 | 1.92 |

*5.3. Statistical Comparisons*

5.3.1. Wilcoxon Signed Rank Test Results

Following the common approach that is applied in the field of PS and PG algorithms [3,4,10,14,24,25,27], the experimental study is complemented with a Wilcoxon signed rank test [36]. Thus, we statistically confirm the validity of the measurements presented in Table 2. The Wilcoxon signed rank test compares all the algorithms in pairs, considering the result achieved against each dataset. We applied the Wilcoxon signed rank test using the PSPP statistical software.

As mentioned above, it is clear that RSP3-M and RSP3E-M compute fewer distances than RSP3-M2 and RSP3E-M2, respectively. Furthermore, RSP3-M2 and RSP3E-M2 compute fewer distances than RSP3-RND and RSP3E-RND, and the latter compute fewer distances than RSP3 and RSP3E. Thus, we do not run the Wilcoxon test for the DC measurements.

Table 4 presents the results of the Wilcoxon signed rank test obtained for the ACC, RR and CPU measurements. The column labeled "w/l/t" lists the number of wins, losses and ties for each comparison test. The column labeled "Wilcoxon" (last column) lists a value that quantifies the significance of the difference between the two algorithms compared. When this value is lower than 0.05, one can claim that the difference is statistically significant.

In terms of accuracy, the results show that the statistical difference between the following pairs is not significant: NOP versus RSP3E, NOP versus RSP3E-RND and NOP versus RSP3E-M2. In contrast, the statistical difference between the conventional RSP3 algorithm and NOP is significant. Thus, we can claim that the 1-NN classifier that runs over the condensing set generated by the proposed RSP3E, RSP3E-RND and RSP3E-M2 algorithms achieves as high accuracy as the 1-NN classifier that runs over the original training set. Moreover, the test shows that there is no significant difference in terms of accuracy between the original version of RSP3 and the following proposed variants: RSP3E, RSP3-RND, and RSP3E-RND.

In contrast, there is statistical difference in terms of Reduction Rates and CPU times. This means that we can obtain as high accuracy as that of the original RSP3 algorithm but with lower computational cost, while the cost of the condensing set construction is lower. Moreover, the test confirms that there is statistical difference in terms of accuracy between the pairs RSP3-M versus RSP3-M2 and RSP3E-M versus RSP3E-M2. Although there is a significant difference in terms of RR and CPU measurements, RSP3-M2 and RSP3E-M2 can be considered better. Last but not least, the test shows that RSP3E, RSP3E-RND, RSP3E-M and RSP3E-M2 dominate RSP3, RSP3-RND, RSP3-M and RSP3-M2, respectively, in terms of reduction rates, while the accuracy and the CPU times are not negatively affected.

**Table 4.** Results of Wilcoxon signed rank test on ACC, RR and CPU measurements.

| Methods | Accuracy | | Reduction Rate | | CPU | |
|---|---|---|---|---|---|---|
| | w/l/t | Wilc. | w/l/t | Wilc. | w/l/t | Wilc. |
| NOP vs. RSP3 | 13/3 | **0.020** | - | - | - | - |
| NOP vs. RSP3E | 8/8 | 0.501 | - | - | - | - |
| NOP vs. RSP3-RND | 13/3 | **0.008** | - | - | - | - |
| NOP vs. RSP3E-RND | 9/7 | 0.877 | - | - | - | - |
| NOP vs. RSP3-M | 15/1 | **0.001** | - | - | - | - |
| NOP vs. RSP3E-M | 14/2 | **0.001** | - | - | - | - |
| NOP vs. RSP3-M2 | 14/2 | **0.002** | - | - | - | - |
| NOP vs. RSP3E-M2 | 9/7 | 0.352 | - | - | - | - |
| RSP3 vs. RSP3E | 7/9 | 0.215 | 0/16 | **0.000** | 6/10 | 0.215 |
| RSP3 vs. RSP3-RND | 9/5 | 0.778 | 15/1 | **0.000** | 1/15 | **0.001** |
| RSP3 vs. RSP3E-RND | 8/8 | 0.469 | 2/14 | **0.000** | 1/15 | **0.001** |
| RSP3 vs. RSP3-M | 15/1 | **0.001** | 0/16 | **0.000** | 0/16 | **0.000** |
| RSP3 vs. RSP3E-M | 13/3 | **0.015** | 0/16 | **0.001** | 0/16 | **0.000** |
| RSP3 vs. RSP3-M2 | 14/2 | **0.001** | 0/16 | **0.007** | 0/16 | **0.000** |
| RSP3 vs. RSP3E-M2 | 9/7 | 0.605 | 0/16 | **0.000** | 0/16 | **0.000** |
| RSP3E vs. RSP3-RND | 11/5 | 0.059 | 16/0 | **0.000** | 1/15 | **0.001** |
| RSP3E vs. RSP3E-RND | 11/4/1 | 0.139 | 10/6 | 0.109 | 1/15 | **0.001** |
| RSP3E vs. RSP3-M | 14/2 | **0.003** | 8/8 | 0.717 | 0/16 | **0.000** |
| RSP3E vs. RSP3E-M | 15/1 | **0.001** | 1/15 | **0.001** | 0/16 | **0.000** |
| RSP3E vs. RSP3-M2 | 12/4 | **0.008** | 12/4 | **0.007** | 0/16 | **0.000** |
| RSP3E vs. RSP3E-M2 | 12/4 | **0.007** | 1/14 | **0.004** | 0/16 | **0.000** |
| RSP3-RND vs. RSP3E-RND | 7/9 | 0.148 | 0/16 | **0.000** | 5/10 | 0.532 |
| RSP3-RND vs. RSP3-M | 14/2 | **0.010** | 0/16 | **0.000** | 0/16 | **0.000** |
| RSP3-RND vs. RSP3E-M | 12/4 | 0.079 | 0/16 | **0.000** | 0/16 | **0.000** |
| RSP3-RND vs. RSP3-M2 | 11/4/1 | **0.036** | 0/16 | **0.000** | 1/15 | **0.001** |
| RSP3-RND vs. RSP3E-M2 | 8/8 | 0.277 | 0/16 | **0.000** | 1/15 | **0.001** |
| RSP3E-RND vs. RSP3-M | 13/3 | **0.013** | 8/8 | 0.877 | 0/16 | **0.000** |
| RSP3E-RND vs. RSP3E-M | 14/2 | **0.010** | 1/15 | **0.001** | 0/16 | **0.000** |
| RSP3E-RND vs. RSP3-M2 | 12/4 | **0.030** | 11/5 | **0.015** | 1/14 | **0.001** |
| RSP3E-RND vs. RSP3E-M2 | 13/3 | **0.049** | 2/14 | **0.005** | 1/14 | **0.005** |
| RSP3-M vs. RSP3E-M | 5/11 | 0.056 | 0/16 | **0.000** | 3/12 | 0.100 |
| RSP3-M vs. RSP3-M2 | 4/12 | **0.006** | 14/2 | **0.001** | 15/0/1 | **0.001** |
| RSP3-M vs. RSP3E-M2 | 4/12 | 0.056 | 4/12 | **0.046** | 14/1/1 | **0.001** |
| RSP3E-M vs. RSP3-M2 | 7/9 | 0.501 | 15/1 | **0.001** | 15/0/1 | **0.001** |
| RSP3E-M vs. RSP3E-M2 | 4/12 | **0.020** | 13/3 | **0.002** | 15/0/1 | **0.001** |
| RSP3-M2 vs. RSP3E-M2 | 4/12 | 0.063 | 0/16 | 0.000 | 6/8/2 | 0.975 |

5.3.2. Friedman Test Results

The non-parametric Friedman test was used in order to rank the algorithms. The test ranks the algorithms for each dataset separately. The best performing algorithm is ranked number 1, the second best is ranked number 2, etc. We used the Friedman test through the PSPP statistical software. The test was run three times, one for each criterion measured. Table 5 presents the results of the Friedman test obtained for the ACC, RR and CPU measurements, respectively.

The Friedman test shows that:

- RSP3E is the most accurate approach. RSP3E-RND, RSP3, RSP3-RND and RSP3E-M2 are the runners-up.
- RSP3E-M and RSP3E-M2 achieve the highest RR measurements. RSP3-M and RSP3E are the runners-up.
- RSP3E-M and RSP3-M are the fastest approaches. RSP3-M2 and RSP3E-M2 are the runners-up.

**Table 5.** Results of Friedman test on ACC, RR and CPU measurements.

| Algorithm | Mean Rank | | |
|:---:|:---:|:---:|:---:|
| | ACC | RR | CPU |
| NOP | 6.94 | - | - |
| RSP3 | 5.94 | 2.06 | 1.5 |
| RSP3E | 6.78 | 5.03 | 1.75 |
| RSP3-RND | 5.16 | 1.06 | 3.34 |
| RSP3E-RND | 6.03 | 4.63 | 3.72 |
| RSP3-M | 2.38 | 5.13 | 7.09 |
| RSP3E-M | 3.13 | 7.63 | 7.72 |
| RSP3-M2 | 3.59 | 3.75 | 5.34 |
| RSP3E-M2 | 5.06 | 6.72 | 5.53 |

## 6. Conclusions

This paper proposed three RSP3 variations that aim at reducing the computational cost involved by the original RSP3 algorithm. All the proposed variations replace the costly task of finding the pair of the furthest instances in a subset by a faster procedure. The first one (RSP3-RND) selects two random instances. The second one (RSP3-M) computes and uses the means of the two most common classes in a subset. The last variation (RSP3-M2) uses the instances that are closer to the means of the two most common classes in a subset.

Moreover, the present paper proposed an editing mechanism for noise removal. The latter does not generate a prototype for each homogeneous subset that contains only one training instance. In effect, this instance is considered noise and is removed. The editing mechanism can be incorporated into any RSP3 algorithm (original RSP3 included). Therefore, in this paper, we developed and tested seven new versions of the original RSP3 PG algorithm (i.e., RSP3E, RSP3-RND, RSP3E-RND, RSP3-M, RSP3E-M, RSP3-M2, RSP3E-M2).

The experimental study as well as the Wilcoxon and Fridman tests revealed that the editing mechanism is quite effective since it removes a high number of irrelevant training instances that do not contribute in classification accuracy. Thus, the reduction rates are improved either with gains or, at least, without loss in accuracy. In addition, the results showed that RSP3-M2 is more effective than RSP3-M. Although the RSP3-RND variation is simple, it is quite accurate. This happens because the RR achieved by RSP3-RND is not very high.

In our future work, we plan to develop data reduction techniques for complex data, such as multi-label data, data in non-metric spaces and data streams.

**Author Contributions:** Author Contributions: Conceptualization, S.O., T.M., G.E. and D.M.; methodology, S.O.,T.M., G.E. and D.M.; software, S.O., T.M., G.E. and D.M.; validation, S.O., T.M., G.E. and D.M.; formal analysis, S.O., T.M., G.E. and D.M.; investigation, S.O., T.M., G.E. and D.M.; resources, S.O., T.M., G.E. and D.M.; data curation, S.O., T.M., G.E. and D.M.; writing—original draft preparation, S.O., T.M., G.E. and D.M.; writing—review and editing, S.O., T.M., G.E. and D.M.; visualisation, S.O., T.M., G.E. and D.M.; supervision, S.O., T.M., G.E. and D.M.; project administration, S.O., T.M., G.E. and D.M. All authors have read and agreed to the published version of the manuscript.

**Funding:** This research received no external funding.

**Data Availability Statement:** Publicly available datasets were analyzed in this study. These data can be found here: https://archive.ics.uci.edu/ml/ and https://sci2s.ugr.es/keel/datasets.php (accessed on 2 October 2022).

**Conflicts of Interest:** The authors declare no conflict of interest.

## Abbreviations

The following abbreviations are used in this manuscript:

| | |
|---|---|
| DRT | Data Reduction Technique |
| PG | Prototype Generation |
| PS | Prototype Selection |
| RSP3 | Reduction by Space Partitioning 3 |
| RSP3E | Reduction by Space Partitioning 3 with Editing |
| RSP3-RND | Reduction by Space Partitioning 3 with Random pairs |
| RSP3E-RND | Reduction by Space Partitioning 3 with Editing and Random pairs |
| RSP3-M | Reduction by Space Partitioning 3 using Means |
| RSP3E-M | Reduction by Space Partitioning 3 with Editing using Means |

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
