# Peer review of "Fast Training Set Size Reduction Using Simple Space Partitioning Algorithms"

_information, doi:10.3390/info13120572_

Round 1

Reviewer 1 Report

The authors present the revised reduction by space partitioning (RSP3) algorithms for reducing the computational costs both in time and hardware. In short, the noise determination strategy may avoid unnecessary calculations led by noises. I think this work is interesting and technically useful, although some minor issues should be addressed before the consideration of acceptance.

1.      The paper structure has to be better organized. For example, the 2nd section of Related Work is too long. The authors introduce too much background information and previous researches. This section should be compressed and incorporated into the Introduction section.

2.      In comparison, the experimental results shown in the 5th section are not substantial enough. The calculation regression trends should be presented via figures, in addition to just tables.

3.      The authors may want to provide possible explanations of the calculation-saving principle.

Author Response

Reviewer #1: The authors present the revised reduction by space partitioning (RSP3) algorithms for reducing the computational costs both in time and hardware. In short, the noise determination strategy may avoid unnecessary calculations led by noises. I think this work is interesting and technically useful, although some minor issues should be addressed before the consideration of acceptance.

1.The paper structure has to be better organized. For example, the 2nd section of Related Work is too long. The authors introduce too much background information and previous researches. This section should be compressed and incorporated into the Introduction section.

Answer 1. By taking into consideration the comments of both reviewers, we decided to not merge the first two Sections of our paper. Section 1 introduces the reader to the concept of Prototype Generation and presents the motivation and the contribution of the paper. Section 2 reviews the recent works in the field of Prototype Generation. Thus, we think that the two Sections present quite different aspects and so they were not merged. However, we added some recent relevant works in Section 2.

2. In comparison, the experimental results shown in the 5th section are not substantial enough. The calculation regression trends should be presented via figures, in addition to just tables.

Answer 2. Done. New figures for the average measurements were added and discussed. We would like to thank the reviewer for this comment. We think that the figures constitute an essential addition.

3. The authors may want to provide possible explanations of the calculation-saving principle.

Answer 3. We mention in the paper that the cost of finding the furthest points in order to divide a subset is the dominant cost of the original RSP3. Our variations avoid this cost, hence the considerably lower computational cost.

Reviewer 2 Report

The article has average merit for the publication. Kindly consider the following suggestions:

- Please include the appropriate research gap in related works.

- Kindly add more discussion on Table 2 until Table 4. More analysis can be done. 

- The references needs to be more up-to-date (at least add a few from 2021-2022). 

- The formatting for caption in Figure 1 is somewhere off.

- Please explain the important parameters in the experimental setup. 

Author Response

Reviewer #2: The article has average merit for the publication. Kindly consider the following suggestions:

1. Please include the appropriate research gap in related works. & The references needs to be more up-to-date (at least add a few from 2021-2022).

Answer 1. Done. We added four recent relevant papers in Section 2.

2. Kindly add more discussion on Table 2 until Table 4. More analysis can be done.

Answer 2. Done. We added discussion for the experimental measurements presented in the Tables. New figures for the average measurements were added and discussed. A new table that presents statistics of the measurements is also added and discussed.

3. The formatting for caption in Figure 1 is somewhere off.

Answer 3. Done.

4. Please explain the important parameters in the experimental setup.

Answer 4. Done. We included a sentence concerning the k parameter value in the Experimental setup.

Round 2

Reviewer 2 Report

Overall, the amendments have been made carefully.